# Finding Balance in Adversity: Nitrate Signaling as the Key to Plant Growth, Resilience, and Stress Response

**DOI:** 10.3390/ijms241914406

**Published:** 2023-09-22

**Authors:** Yancong Jia, Debin Qin, Yulu Zheng, Yang Wang

**Affiliations:** 1State Key Laboratory of Plant Environmental Resilience, College of Biological Sciences, China Agricultural University, Beijing 100193, China; s20213020236@cau.edu.cn; 2State Key Laboratory of Protein and Plant Gene Research, School of Life Sciences, Peking University, Beijing 100871, China; qindebin@pku.edu.cn; 3College of Biological Sciences, China Agricultural University, Beijing 100193, China; 2021302040312@cau.edu.cn

**Keywords:** nitrate signaling, primary nitrate response, abiotic stresses, NRT1.1, NLP7, CIPK23, epigenetic regulators

## Abstract

To effectively adapt to changing environments, plants must maintain a delicate balance between growth and resistance or tolerance to various stresses. Nitrate, a significant inorganic nitrogen source in soils, not only acts as an essential nutrient but also functions as a critical signaling molecule that regulates multiple aspects of plant growth and development. In recent years, substantial advancements have been made in understanding nitrate sensing, calcium-dependent nitrate signal transmission, and nitrate-induced transcriptional cascades. Mounting evidence suggests that the primary response to nitrate is influenced by environmental conditions, while nitrate availability plays a pivotal role in stress tolerance responses. Therefore, this review aims to provide an overview of the transcriptional and post-transcriptional regulation of key components in the nitrate signaling pathway, namely, NRT1.1, NLP7, and CIPK23, under abiotic stresses. Additionally, we discuss the specificity of nitrate sensing and signaling as well as the involvement of epigenetic regulators. A comprehensive understanding of the integration between nitrate signaling transduction and abiotic stress responses is crucial for developing future crops with enhanced nitrogen-use efficiency and heightened resilience.

## 1. Introduction

Plants possess a remarkable ability to sense and respond to changes in their environment, ensuring their survival and successful reproduction. To adapt and withstand various abiotic and biotic stresses, plants must strike a delicate balance between growth and defense. Nitrate, a major inorganic nitrogen source in soils, serves not only as an essential nutrient but also as a crucial signaling molecule that regulates multiple aspects of plant growth and development, including seed germination, root system architecture, flowering, and the coordinated absorption of other mineral nutrients [1]. Here, the current research progress on nitrate signaling and its potential roles in responding to abiotic stresses, such as drought, salinity, high temperature, and cold stress, will be discussed.

## 2. Abiotic Stresses and Plant Responses

Plants are constantly exposed to a wide range of abiotic stresses, which can have detrimental effects on plant growth, development, and productivity [2]. Plants adapt to water scarcity by closing stomata, rolling leaves, and adjusting osmotic balance. They activate antioxidant defense systems and utilize phytohormones like abscisic acid (ABA) for drought stress signaling. When plants are exposed to high salt concentrations in the soil, they experience both osmotic stress and ion toxicity. Plants respond by adjusting osmotic balance through the accumulation of compatible solutes and regulating ion homeostasis through exclusion, compartmentalization, and selective uptake of ions. Salinity stress triggers a complex signaling network involving calcium, reactive oxygen species (ROS), and ABA. In response to cold stress (chilling or freezing), plants accumulate cryoprotectants (such as sugars and proteins) that help protect cellular structures, maintain osmotic balance, and induce cold-responsive genes that encode proteins like cold shock proteins, antifreeze proteins, and dehydrins. Similarly, high-temperature stress (heat stress) induces the synthesis of heat shock proteins, activates antioxidant defenses, and increases membrane stability. Therefore, some abiotic stress responses are common (such as ABA signaling under drought and salinity stresses), while others are stress-specific (such as ion compartmentalization induced by salinity stress). Abiotic stresses are detected within different cellular compartments, triggering molecular responses at multiple levels with interconnected signaling pathways [3].

## 3. Overview of Nitrate Signaling Pathway

Nitrate, a significant inorganic nitrogen source in soils, not only acts as an essential nutrient but also functions as a critical signaling molecule that regulates multiple aspects of plant growth and development (Figure 1). For over two decades, the dual-affinity nitrate transporter NRT1.1 (CHL1/NPF6.3) in *Arabidopsis* has been established as a nitrate sensor or transceptor that plays a critical role in nitrate signaling, also known as primary nitrate responses (PNRs). In the presence of nitrate treatments, NRT1.1 triggers the rapid induction of nitrate assimilation genes and nitrate transporter genes, orchestrating the plant’s response to efficiently uptake and utilize nitrate as a nutrient source [4]. Nitrate provision results in a transient increase in the cytoplasmic calcium concentration, forming a nitrate-triggered specific calcium signal. It has been shown that inositol 1, 4, 5-triphosphate (IP_3_) concentrations increased together with calcium concentration in response to nitrate treatments, indicating that PHOSPHOLIPASE C (PLC) activity takes part in this process [5]. Recent studies have revealed new insights into the role of CNGC15, a cyclic nucleotide-gated channel with calcium channel activity, in generating a nitrate-specific calcium signature in *Arabidopsis*. NRT1.1 forms a transporter complex with CNGC15 on the cell membrane, inhibiting its calcium channel activity. When the external nitrate concentration increases, the interaction between NRT1.1 and CNGC15 weakens, allowing the NRT1.1–CNGC15 complex to regain its calcium channel activity, thereby forming a dynamic gate under the regulation of nitrate [6].

The nitrate-triggered cytoplasmic calcium waves could be sensed by CALCINEURIN-B LIKE PROTEIN 1/9 (CBL1/9), which interacts with and activates CBL-INTERACTING PROTEIN KINASE 23 (CIPK23) under low-nitrate concentration conditions [7,8,9]. A protein phosphatase ABA INSENSITIVE 2 (ABI2) is reported to dephosphorylate CBL1 and CIPK23, thereby inactivating this calcium sensor/kinase complex and preventing NRT1.1 phosphorylation [7]. The activity of ABI2 could be inhibited by ABA [10]. CIPK23 phosphorylates NRT1.1 at its Thr101 residue, switching the transporter from low-affinity (K_M_ ~ 4 mM) to high-affinity (K_M_ ~ 40 μM) mode [4]. Meanwhile, the phosphorylated and non-phosphorylated forms of NRT1.1 probably have distinct signaling roles. The nitrate-triggered transient accumulation of calcium in the cytoplasm of root cells is much greater in plants expressing the non-phosphorylatable NRT1.1^T101A^ mutant than in those expressing the phosphomimic NRT1.1^T101D^ mutant [11]. Indeed, the phosphomimic NRT1.1^T101D^ mutant is able to activate PNR and the nitrate-dependent regulation of root development, whereas the non-phosphorylatable NRT1.1^T101A^ mutant fails partially [11,12]. Another member of CIPKs, CIPK8, is found to positively regulate the low-affinity nitrate response. The CBL protein partner and direct target(s) of CIPK8 remain to be revealed.

CALCIUM-DEPENDENT PROTEIN KINASE (CPK) proteins play a key role in the nitrate signaling pathway, linking the nitrate sensing in the plasma membrane with nuclear regulators, NIN-LIKE PROTEIN (NLP) transcription factors. Evidence shows that nitrate induces calcium accumulation in the nucleus and some CPKs translocation to the nucleus [13]. The mechanism by which CPK10/30/32 are translocated from the cytoplasm to the nucleus in response to nitrate, as well as how the increase in nuclear calcium concentration is linked to cytoplasmic calcium waves, remains poorly understood and requires further investigation. The activated CPK10/30/32 phosphorylate NLP7 (and NLP6) at the conserved Ser205 residue to ensure its nuclear retention, which further leads to a downstream nitrate response [13]. A different phosphorylation modification status of NLP7 mediated by SNF1-RELATED KINASE 1 (SnRK1) could suppress its activity. Under carbon deficiency, the α-catalytic subunit of the energy sensor SnRK1, KIN10, phosphorylates NLP7 at Ser125 and Ser306, which increases the cytoplasmic localization and promotes the subsequent degradation of NLP7. Nitrate treatment could release the inhibitory effects of KIN10 by promoting its degradation [14].

As the master regulator of PNR, NLP7 binds to the promoters of 851 genes, when nitrate is present. Many transcription factor genes are also regulated directly by NLP7 [13,15,16]. These nitrate-response secondary transcription factor targets of NLP7, including LOB domain-containing protein 37/38/39 (LBD37/38/39) [17], NITRATE-INDUCIBLE GARP-TYPE TRANSCRIPTIONAL REPRESSOR 1/HYPERSENSITIVITY TO LOW PI-ELICITED PRIMARY ROOT SHORTENING 1/HRS1 HOMOLOG 1 (NIGT1/HRS1/HHO1) [18,19], TGACG-BINDING FACTOR 1/4 (TGA1/4) [20], CELL GROWTH DEFECT FACTOR 1 (CDF1) [21], NUCLEAR FACTOR Y, SUBUNIT A3 (NF-YA3/HAP2C) [15], HOMEOBOX PROTEIN 52/54 (HB52/54) [22], and HOMOLOG OF BRASSINOSTEROID ENHANCED EXPRESSION2 INTERACTING WITH IBH1 (HBI1) [23], amplify the NLP7-initiated transcriptional cascade for modulation of various processes, such as root development, chloroplast development, phosphate homeostasis, carbohydrate metabolism, and amino acid metabolism. The transcription factor HBIs not only regulate PNR gene expression but also increase the expression levels of a set of antioxidant genes to reduce the accumulation of H_2_O_2_, which inhibits the nitrate-induced nuclear localization of NLP6 and NLP7, thereby forming a feedback regulatory loop to enhance the nitrate signaling [17,20,21,23].

ARABIDOPSIS NITRATE REGULATED 1 (ANR1) is a transcription factor which is responsible for lateral root elongation in the nitrogen-rich condition [24]. ANR1 is induced from the endosome by the Ca^2+^–CPKs–NLPs signaling pathway under high nitrate conditions [11]. Some transcription factors have direct interaction with NLP7 in the nucleus, such as NITRATE REGULATORY GENE 2 (NRG2) and TEOSINTE BRANCHED 1/CYCLOIDEA/PROLIFERATING CELL FACTOR 1-20 (TCP20). NRG2 positively modulates *NRT1.1* expression and physically interacts with NLP7 to be involved in PNRs [25]. TCP20 interacts with NLP6/7 under both continuous nitrate supply and N-starvation conditions [26]. Under nitrogen starvation, TCP20-NLP6/7 heterodimers accumulate in the nucleus, induce the expression of PNR genes, and repress the expression of a cell-cycle progression gene, *CYCLIN-DEPENDENT PROTEIN KINASE CYCB1;1* [26]. Meanwhile, TCP20 positively regulates the short mobile peptides C-TERMINALLY ENCODED PEPTIDES (CEPs)-mediated systemic nitrate acquisition [27,28].

Recently, the NLP7 protein has been reported to have an additional sensor function, which makes nitrate sensing and signaling more complicated. Nitrate could directly bind to NLP7 at its N terminus to derepress its transcriptional activation activity [29]. What is more, evidence shows that the nine members of *Arabidopsis* NLPs can form homo- or heterodimeric complexes, through the protein interaction PB1 domain, to regulate PNR with functional redundancy or gene-specific regulatory nature [30,31,32]. NLP7 and NLP6 have interactions with each other and display partial functional redundancy in regulating nitrate-induced genes, with NLP7 exerting a broader impact than NLP6. They are completely functionally redundant in negatively regulating ammonium-responsive genes when ammonium is the sole nitrogen source [32]. NLP2 also interacts with NLP7 in vivo and shares some target genes with NLP7, linking nitrate assimilation with carbon metabolism for efficient nitrogen use and biomass production [33].

Research on nitrate signaling in other crops is still in its early stages. In rice, a distinct OsNRT1.1B–OsSPX4–OsNLP3 module is discovered as a nitrate signaling transduction pathway. OsNRT1.1B, the postulated orthologue of the *Arabidopsis* NRT1.1 transceptor, interacts with the phosphate signaling repressor protein OsSPX4 (SPX DOMAIN GENE 4), which suppresses the activity of the master transcription factor OsNLP3. Once nitrate is sensed by OsNRT1.1B, OsNBIP1 (NRT1.1B INTERACTING PROTEIN 1)—an E3 ubiquitin ligase, is recruited to OsNRT1.1B–OsSPX4 complex to mediate the ubiquitination and degradation of OsSPX4. Hence, OsNLP3 is released from interaction with OsSPX4 and translocates to the nucleus to trigger the nitrate response [34,35,36]. It is still unknown whether SPXs–NLPs in *Arabidopsis* also integrate phosphate and nitrate signaling. Much less is known in maize. ZmNLP3.1 recognizes nitrate-responsive cis-elements (NREs) in the promoters of the *ZmNRT2.1* and *ZmNRT2.2* genes, and it activates the expression of these genes in response to nitrate [37].

## 4. Regulations of NRT1.1 under Abiotic Stresses

NRT1.1 is a versatile protein located on the plasma membrane, serving multiple functions. It functions as the main nitrate transporter, responsible for the majority of nitrate uptake required for plant growth [1,38]. The transport of nitrate by NRT1.1 is coupled with protons, leading to changes in intracellular and extracellular pH levels [39]. In addition, NRT1.1 also facilitates the transport of auxin, a plant hormone involved in growth regulation. The transportation of auxin by NRT1.1 is hindered by nitrate in a dose-dependent manner. Therefore, NRT1.1 acts as a regulatory switch that integrates both nitrate signaling/transport and auxin signaling/transport to govern plant growth and developmental processes [40]. Furthermore, NRT1.1 is also involved in chloride transport, leading to the accumulation of chloride ions [41].

NRT1.1 has been reported extensively involved in drought and salt stress sensitivity, ammonium/low pH tolerance, and heavy metals resistance with different mechanisms [11,40,41,42,43,44,45,46,47,48]. Research progress on NRT1.1’s roles in abiotic stress resistance up to 2021 could be referred to in an excellent published review [49]. In general, the transcriptional expression of *NRT1.1* is controlled by transcription factors, while the transport activity of NRT1.1 can be influenced by kinases and phosphatases in response to abiotic stresses. Under low rhizosphere pH, the transcription factor SENSITIVE TO PROTON RHIZOTOXICITY 1 (STOP1) directly activates the transcription of *NRT1.1*, enhancing the coupled proton-driven nitrate uptake. This process promotes proton consumption in the rhizosphere, increases rhizosphere pH, and alleviates proton toxicity in plants [50]. In response to drought stress, the BASIC LEUCINE ZIPPER 62 (bZIP62) transcription factor regulates the transcription of *NRT1.1* [51]. Drought stress induces CPK6-mediated phosphorylation of NRT1.1 at Thr447 to repress its nitrate transport activity [52]. SnRK2.2/2.3/2.6 kinase proteins interact with and phosphorylate NRT1.1 at Ser585 to impair its nitrate transport activity as an effect of stress-response hormone ABA signaling [53]. However, the precise mechanisms underlying the nitrate sensing function of NRT1.1 in relation to the adaptive response to abiotic stress are still not well understood. It is likely that alterations in the protein level of NRT1.1 on the plasma membrane could impact nitrate binding and signal perception. Furthermore, protein modifications at specific amino acid residues that influence nitrate transport may also have additional effects on nitrate binding. More robust experimental evidence is needed to further explore and clarify these aspects. It is well known that NRT1.1-mediated nitrate signaling is tightly associated with Thr101 phosphorylation by CBL1/9-CIPK23 module [4,54]. ABI2 is able to dephosphorylate CBL1 and CIPK23 to inhibit nitrate signaling by preventing the NRT1.1 phosphorylation [7]. This is a potentially functional link between nitrate signaling and abiotic stress response. Upon stress exposure (for example, drought, salinity and osmotic stress), ABA accumulation would inactivate ABI2 and thereby promote the activity of CBL1-CIPK23 to enhance NRT1.1-mediated nitrate signaling.

## 5. Calcium Sensors as Regulating Hubs in Stress Tolerance

There are four major families of calcium sensors in plants: CALMODULINs (CaMs), CALMODULIN-LIKE proteins (CMLs), CBLs, and CPKs. Unlike CaMs, CMLs, and CBLs that must relay Ca^2+^-induced conformational changes to their partner proteins for signaling, CPKs harbor four EF-hand Ca^2+^-binding motifs and a Ser/Thr kinase domain, which brings together both Ca^2+^ sensing and responding activity within a single protein [55]. Up to date, CBL1/9 and its partner protein, CIPK23 and CPK10/30/32 have been reported to participate in the nitrate signaling [4,13].

CBL1/9–CIPK23 plays a dual role in nitrate signaling. On the one hand, CBL1/9–CIPK23 is the downstream of nitrate–NRT1.1–CNGC15–calcium waves. On the other hand, CBL1/9–CIPK23 targets and phosphorylates NRT1.1 at Thr101 to regulate NRT1.1-dependent nitrate transport and signaling. Besides NRT1.1, many ionic transporters are direct targets of CIPK23, including ARABIDOPSIS K TRANSPORTER 1 (AKT1) [56], HIGH-AFFINITY K TRANSPORTER 5 (HAK5) [57], TINY ROOT HAIR 1 (TRH1/KUP4) [58], AMMONIUM TRANSPORTER 1;1/1;2 (AMT1;1/1;2) [59], FERRIC REDUCTASE OXIDASE 2 (FRO2) [60], IRON REGULATED TRANSPORTER 1 (IRT1) [61], SLOW ANION CHANNEL-ASSOCIATED 1 (SLAC1), and manganese transporter NATURAL RESISTANCE-ASSOCIATED MACROPHAGE PROTEIN 1 (NRAMP1) [62,63]. STOP1 regulates salt and drought tolerance by transcriptionally induced *CIPK23* [64]. Over-expression of *TaCIPK23* rendered ABA sensitivity in wheat, as evidenced by delayed seed germination and the induction of stomatal closure [65]. Research progresses on the prominent roles of CIPK23 in the regulation of plant nutrient transporters up to 2021 could be referred to in these excellent published reviews [66,67,68]. Low nitrate will lead to CIPK23 activation to target NRT1.1, while low potassium and iron deficiency could also evoke an increase in cytosolic calcium concentration and activate CBL1/9–CIPK23. It will be interesting to test whether the calcium signature has some specificity under various stresses and whether the stress-induced activation of CBL1/9–CIPK23 would influence NRT1.1-mediated nitrate signaling by phosphorylating NRT1.1 at the same time. Moreover, the CBL9–CIPK23 complex is inhibited by the dimer coupling state of NRT1.1 at high nitrate concentrations [9], which means that it might also influence the transport of other ions or the responses to certain stresses. During high external ammonium concentrations (such as ammonium toxicity), elevated ABA level is sensed by ABI1–PYR1-like complexes followed by the inactivation of phosphatase ABI1, in turn activating CIPK23 for AMTs phosphorylation and ammonium transport activity inhibition [69]. In another study, ABA accumulation inactivates phosphatase ABI2 and activates CIPK23, thereby enhancing the phosphorylation of NRT1.1, resulting in a reduced nitrate uptake [7]. Therefore, it seems that CIPK23 acts as a regulating hub in diverse abiotic stresses and nitrate signaling process. The underlying mechanisms still need more work to elucidate.

CPK10, CPK30, and CPK32 are important players in transducing the nitrate-triggered calcium signature in the nucleus and activating NLP7 to modulate transcriptional responses to nitrate availability [13]. CPK family members have been found to have diverse functions in response to abiotic and biotic stresses [70]. CPK10 plays important roles in ABA- and calcium-mediated regulation of stomatal movements in response to drought stress [71]. *CPK30* was extensively up-regulated in salt-stressed seedlings, suggesting that it might play important roles in salt tolerance [72]. CPK32 is able to phosphorylate the ABA-responsive transcription factor ABF4, and CPK32-overexpressing plants showed a hypersensitive phenotype to ABA [73]. CPK32 is also responsible for the ammonium uptake activity of AMT1.1 [74]. Therefore, CPK10/30/32 might have distinct roles in signaling integration.

In 2021, a calmodulin-like protein, CALMODULIN-LIKE-38 (CML38), is reported to regulate root elongation in response to low nitrate. CML38 physically interacts with PEP1 RECEPTOR 2 (PEPR2), participating in the BR signal transduction promoted by low nitrate. Nitrate enhances the transcriptional levels of *CML38* in Arabidopsis roots. After the nitrate treatment, the transcription levels of *NRT1.1*, *NITRATE REDUCTASE* (*NR*), and *NITRITE REDUCTASE* (*NiR*) increase significantly higher in *cml38* mutant than in wild-type plants. The expression levels of *NRT2.1*, *NR*, and *NiR* are enhanced in the *pepr2* mutant. The functional association of CML38 and PEPR2 in nitrate signaling is independent of the Pep1–PEPR2 pathway, though PEPR2 is a Pep1 peptide receptor required for the perception of Pep peptides and defense responses [75]. The transcript of *CML38* is also up-regulated more than 300-fold in roots within 6 h of hypoxia treatment. By using yellow fluorescent protein (YFP) translational fusions, CML38 protein is found to be localized to messenger RNA ribonucleoprotein (mRNP) complexes, including stress granules [76]. CML38 regulates hypoxia-induced stress granule turnover by autophagy [77]. Therefore, it is highly plausible that CML38 plays a role in integrating nitrate signaling with hypoxia stresses, such as flooding.

## 6. Roles of NLP7 in Abiotic Stresses

NLP7 is the master regulator controlling the primary response in nitrate signaling. The loss-of-function *nlp7* exhibits a nitrogen-deficient phenotype as well as tolerance to drought, salt stress [78], and freezing sensitivity [79]. Under salt stress, the *nlp7* plants could accumulate more Na^+^ and K^+^ and less ABA than wild-type plants. The transcript level of *NLP7* is increased in response to salt stress [78]. Cold stimulates the nuclear translocation of NLP7 after phosphorylation by CPK28 at Ser783, Ser793, Ser807, Ser808, and Thr817 for cold stress response. Overexpression of NLP7^5A^ (harboring mutations of CPK28 phosphorylation sites) failed to rescue the freezing sensitivity of *nlp7* but fully rescued nitrate sensitivity. By contrast, the overexpression of NLP7^S205A^ (carrying a mutation of CPK10/30/32 phosphorylation site) failed to restore the nitrate sensitivity of *nlp7* but fully restored freezing tolerance. These results indicate that phosphor-regulation of NLP7 by different CPKs is responsible for distinct signaling pathways. However, RNA-seq data suggest that although the target genes of nitrate-activated and cold stress-activated NLP7 are quite different, there are some common target genes of NLP7 in responses to nitrate and cold stress, which implies a possible link between nitrogen and cold signaling [79].

The nucleic localization and protein stability are important for NLP7 to activate downstream gene transcription. Nitrate-induced calcium signature activates CPK10/30/32 that phosphorylate NLP6/7 at Ser205 to ensure their nucleic localization for transcriptional activation of target genes [13]. Another kinase, SnRK1, phosphorylates NLP7 at Ser125 and Ser306, which increases the cytoplasmic localization and promotes the subsequent degradation of NLP7 [14]. SnRK1 is activated by nutrient deficiency and plant stress response to salt stress, oxidative stress, and ER stress [80,81]. This suggests that adverse conditions may potentially influence NLP7 at the protein level through SnRK1, thereby regulating nitrate signaling. ROS homeostasis affects many developmental processes and responses to diverse abiotic stresses [82]. The nucleocytoplasmic shuttling of NLP7 is inhibited by H_2_O_2_. Nitrate treatment reduced the accumulation of H_2_O_2_, through HBI-mediated ROS homeostasis. Mutation in *HBI* genes resulted in ROS accumulation, and about 22% of nitrate-responsive genes were no longer regulated by nitrate [23]. Nitrate is also the main source for plants to produce nitric oxide (NO), which regulates growth and stress responses [83]. Evidence shows that NLP7 could be targeted by the N-degron proteolytic pathway to regulate the nucleocytoplasmic translocation of NLP7. Protein modification linked to NO action was predicted to occur in NLP7, including the nitration of Tyr157 and Tyr288 as well as the S-nitrosation of Cys2 and Cys374 [84]. The underlying mechanism still needs to be explored.

## 7. Tissue-Specific Nitrate Signaling in Abiotic Stresses

Nitrate serves as a crucial nitrogen source for plants in the soil. Once taken up by the roots, nitrate can undergo assimilation, be stored in vacuoles, or primarily transported through the xylem to the above-ground tissues. To meet the nutrient demands of young leaves, nitrate can be remobilized from older or mature leaves via the phloem. Additionally, seeds also have the ability to store nitrate. It is important to note that the response to nitrate is specific to individual cells and tissues within the plant [85,86]. A high-resolution spatiotemporal transcriptional map of the nitrate response in the Arabidopsis root has been present in 2022. The results show that endodermis is the cell type enriched in regulatory interactions and nitrate cross-talks with ABA mediated by ABSCISIC ACID RESPONSIVE ELEMENTS-BINDING FACTOR 2/3 (ABF2/3) in the endodermis for lateral root growth in response to nitrate [87]. In 2023, CNGC15 is reported to be nuclear localized in 6-day-old *Arabidopsis* roots to modulate the nuclear calcium release associated with root meristem development, indicating nitrate-induced gene expression in *Arabidopsis* root tips before 6 days after germination would be independent of CNGC15. At later stages of root growth, CNCG15 relocalizes to the plasma membrane specifically in columella cells upon high nitrate treatment [88]. Therefore, nitrate signaling transduction is specific to cell types and developmental stages. It will be interesting to figure out whether nitrate sensing is tissue-specific and subcellular localization-specific.

NRT1.1 proved to be a nitrate transceptor in 2009 through the uncoupling of transport and signaling functions of the protein [4]. NRT1.1 is mainly expressed in the epidermis–cortex and central cylinder of mature roots [89]. As a membrane protein, nitrate in the soil is sensed by NRT1.1 to regulate root architecture and seed germination. In 2021, NRT1.13, which shares 37% sequence identity with NRT1.1, has been reported as a potential nitrate sensor (or at least part of a transceptor complex) in the xylem parenchyma cells and regulates shoot architecture and flowering time [90]. When plants are grown under low concentrations of nitrate conditions, *nrt1.13* mutants show reduced lateral nitrate allocation to nodes, increased node number, and delayed flowering. NRT1.13 shows the ability to bind nitrate but has no transport activity in the oocyte experimental system. Moreover, NLP7 is proven to be an intracellular nitrate sensor [29]. The *NLP7* gene is expressed in all plant organs. Because the nitrate binding site of NLP7 is evolutionarily conserved, there probably be more NLP orthologs acting as nitrate sensors. Now researchers have found that all nine Arabidopsis NLPs contribute to nitrate response with unique and redundant roles [31]. Both NLP2 and NLP7 were shown to play major roles in vegetative growth, with a specific function of NLP2 in linking nitrogen and carbon metabolism [33]. NLP6 displays partial functional redundancy in regulating nitrate-induced genes and completes redundancy in negatively regulating ammonium-responsive genes [32]. NLP8 functions in nitrate-promoted seed germination [91]. It is still unknown whether NLP8 functions depend on NLP7. The NLP members can physically interact with each other or with other transcription factors through their C-terminal PB1 domain. These interactions may play specific roles in cell-type-specific and tissue-specific nitrate signaling and produce a broad range of outcomes to cope with developmental and environmental demands. Future studies should focus on the specific nitrate response in different cells/tissues for the regulation of plant tolerance to certain environmental stresses.

## 8. Epigenetic Regulators Integrate Nitrate Signaling and Abiotic Stress Response

Epigenetic mechanisms such as DNA methylation, histone modifications, histone variants, small RNAs, and long noncoding RNAs (lncRNAs) can regulate the expression of genes. The dynamics of epigenetic codes play important roles in regulating genes in response to environmental stresses [92,93]. In 2011, researchers reported that nitrate uptake is under systemic feedback repression by the N satiety of plants involving chromatin remodeling. Repression of *NRT2.1* transcription by high-concentration nitrogen supply is associated with a HIGH NITROGEN INSENSITIVE 9 (HNI9)-dependent increase in histone H3 lysine 27 trimethylation (H3K27me3) at the *NRT2.1* locus [94]. The histone methyltransferase SET DOMAIN GROUP8 (SDG8) mediates genome-wide changes in histone H3 lysine 36 trimethylation (H3K36me3) after nitrate treatments, affecting multiple gene regulatory processes, including genome-wide histone modification, transcriptional regulation, and RNA processing [95]. The function of SDG8 in mediating nitrate responses is conserved across species. Two tomato homologs of SDG8, SlSDG33 and SlSDG34, are also shown to affect H3K4 and H3K36 methylation. In response to nitrate, SlSDG33 and SlSDG34 mediate gene regulation in an organ-specific manner: In roots, SlSDG33 and SlSDG34 regulate a gene network including *NRT1.1* and *Small Auxin Up-regulated RNA* (*SAUR*) genes, as well as N-responsive root growth; in shoots, SlSDG33 and SlSDG34 affect the expression of photosynthesis genes and photosynthetic parameters [96]. A multi-omics study in 2019 provides direct evidence that chromatin accessibility is a determinant of the rapid nitrate response [97]. In maize, ZmCHB101 is the core subunit of the SWI/SNF-type ATP-dependent chromatin remodeling complex. In the presence of nitrate, the binding affinity of ZmCHB101 for NREs decreases dramatically, leading to reduced nucleosome density at NREs and consequently increased ZmNLP3.1 binding to activate the expression of *ZmNRT2.1* and *ZmNRT2.2* genes [37]. ZmCHB101 plays an essential role in leaf development and dehydration and abscisic acid responses [98,99], which implies that chromatin regulators integrate abiotic stress adaptation and nitrate signaling. And major genes play a role in integrating nitrate signaling and abiotic stress responses in Table 1.

Several regulatory RNAs have been reported to be involved in nitrate signaling. Nitrate specifically induces expression of *MIR393* and its target *AUXIN SIGNALING F-BOX 3* (*AFB3*) to regulate auxin-related root architecture in response to nitrate [100]. The expression of *MIR393* is induced by salt stress, heavy metal stresses, and drought stress and repressed by low temperature. *MIR393* and its target genes are strongly conserved in plants [101,102,103]. Similarly, nitrogen metabolites repress levels of *MIR167* to permit the *AUXIN RESPONSE FACTOR 8* (*ARF8*) transcript to accumulate in the pericycle regulating lateral root growth [85]. In rice, *OsMIR444*, which is specific to monocots, targets four genes that are homologous to *ANR1*. *OsMIR444* plays multiple roles in the rice nitrate signaling pathway in nitrate-dependent root growth and nitrate accumulation [104]. *OsMADS27* is identified as the major *OsMIR444* target that regulates the expression of nitrate transporters and root development genes [105]. The expression of *OsMADS27* is induced by nitrate and salt stress. OsMADS27 directly binds to the promoters of *OsHKT1.1* and *OsSPL7* to regulate their expression in response to salt stress. OsMADS27-mediated salt tolerance is nitrate-dependent and positively correlated with nitrate concentration [106]. Therefore, the *MIR393*/AFBs and *OsMIR444*/OsMADS27 modules are the potential links between nitrate signaling and abiotic stress response. The combination of RNA-seq and sRNA-seq analysis has been employed to identify additional miRNA/target modules in nitrate signaling. Through this research approach, *MIR5640* and its target gene *PHOSPHOENOLPYRUVATE CARBOXYLASE 3* (*PPC3*) have been discovered to play a role in regulating the carbon flux necessary for the assimilation of nitrate into amino acids [107]. Using similar experimental strategies, the miRNA-target pairs *ptc-miR169i/b-D6PKL2*, *ptc-miR393a-5p-AFB2*, *ptc-miR6445a-NAC14*, *ptc-miR172d-AP2*, *csi-miR396a-5p_R+1_1ss21GA-EBP1*, *ath-miR396b-5p_R+1-TPR4*, and *ptc-miR166a/b/c-ATHB-8* are found in poplar plants, contributing to root morphology changes in nitrate/ammonium treatments [108].

LncRNAs are a type of less conserved RNA molecules of more than 200 nucleotides (nt) in length. LncRNAs have regulatory roles in transcriptional, posttranscriptional, and epigenetic levels by interacting with DNAs, RNAs, and proteins. Transcript abundances of lncRNAs in *Arabidopsis* exhibit diverse patterns in different tissues in response to nitrate treatment. Among them, *T5120* is strongly induced by nitrate in roots and acts as downstream of NRT1.1 and NLP7 in nitrate signaling to regulate the expression of *NIA1*, *NIA2*, and *NIR*. The underlying mechanism of *T5120* is still unclear [109]. More and more data suggest that lncRNAs play an important role in plant growth and development, as well as stress adaptation. Research progress on lncRNAs in plants up to 2022 could be referred to in these excellent published reviews [93,110]. In the future, more epigenetic regulators need to be investigated to gain a comprehensive understanding of nitrate signaling.

**Table 1 ijms-24-14406-t001:** Major genes play a role in integrating nitrate signaling and abiotic stress responses.

Species	Gene Name	Abiotic Stress	Effects	References
*Arabidopsis thaliana*	*NRT1.1*	Drought stress	The nitrate transport activity of NRT1.1 is repressed under drought, leading to smaller stomatal opening.	[44,51,52]
		Salt stress	The up-regulation of NRT1.1 mediated Cl^−^ uptake under NH_4_^+^-aggravated salt stress.	[41]
		Ammonium toxicity	Ammonium toxicity is related to the nitrate-independent signaling function of NRT1.1. Mutation of NRT1.1 enhances ammonium tolerance.	[45]
		Low pH	H^+^ toxicity induces STOP1 to activate the transcription of *NRT1.1*, enhancing the coupled proton-driven nitrate uptake, which increases rhizosphere pH and alleviates proton toxicity.	[50]
		Iron deficiency	Transcript level of NRT1.1 is down-regulated by Fe^−^deficiency stress, which helps plants better adapt to Fe-deficiency stress.	[47]
	*STOP1*	Drought stress	STOP1 suppresses drought tolerance by regulating K^+^ transport.	[111]
	*SnRK2.2/2.3/2.6*	ABA homeostasis	SnRK2.2/2.3/2.6 kinase proteins phosphorylate NRT1.1 at Ser585 to impair its nitrate transport activity as an effect of stress-response hormone ABA signaling.	[53]
	*ABI1*	Ammonium toxicity	The inactivation of phosphatase ABI1 under high external ammonium concentrations, in turn, activates CIPK23 for AMT phosphorylation and ammonium transport activity inhibition.	[69]
	*ABI2*	ABA homeostasis	ABA accumulation would inactivate ABI2 and thereby promote the activity of CBL1-CIPK23 to enhance NRT1.1-mediated nitrate signaling.	[7]
	*CIPK23*	Drought stress	CIPK23 regulates drought tolerance by combining with CBL1 and CBL9. The complexes can change the ABA sensitivity in guard cells.	[112]
		Salt stress	STOP1 regulates salt and drought tolerance by transcriptionally induced CIPK23.	[64]
		Ammonium toxicity	During high external ammonium concentrations, activating CIPK23 for AMTs phosphorylation and ammonium transport activity inhibition.	[69]
		Low-K^+^	CIPK23 phosphorylates AKT1 transporters that enhance K^+^ uptake.	[66]
	*CPK10*	Drought stress	CPK10 plays important roles in ABA- and calcium-mediated regulation of stomatal movements in response to drought stress.	[71]
	*CPK30*	Salt stress	*CPK30* was up-regulated in salt-stressed seedlings, suggesting that they might play important roles in salt tolerance.	[72]
	*CPK32*	ABA homeostasis	CPK32 is able to phosphorylate the ABA-responsive transcription factor ABF4, and CPK32-overexpressing plants showed a hypersensitive phenotype to ABA.	[73]
	*CML38*	Hypoxia stress	CML38 regulates hypoxia-induced stress granules turnover by autophagy.	[77]
	*NLP7*	Drought stress	The loss-of-function *nlp7* exhibits a nitrogen-deficient phenotype as well as tolerance to drought.	[78]
		Salt stress	The *nlp7* exhibits a nitrogen-deficient phenotype as well as tolerance to salt.	[78]
		Cold stress	Cold stimulates the nuclear translocation of NLP7 after phosphorylation by CPK28 for cold stress response.	[79]
	*SnRK1*	Cold stress	SnRK1 and TOR with ATPase act together to regulate the energy homeostasis under cold stress.	[81]
	*HBI*	ROS homeostasis	Nitrate treatment reduced the accumulation of H_2_O_2_, through HBI-mediated ROS homeostasis.	[23]
*Triticum aestivum*	*TaCIPK23*	Drought stress	Wheat and Arabidopsis overexpressing TaCIPK23 showed a higher survival rate under drought conditions with enhanced germination rate.	[65]
*Zea mays*	*ZmCHB101*	Osmotic stress	ZmCHB101 affects gene expression by remodeling chromatin states and controls RNAPII occupancies in maize under osmotic stress.	[99]
*Oryza sativa*	*OsMADS27*	Salt stress	OsMADS27 directly binds to the promoters of OsHKT1.1 and OsSPL7 to regulate their expression in response to salt stress.	[106]

## 9. Perspectives

Nitrate serves not only as a vital nutritional element but also as a crucial signaling molecule in plant growth and development. These days, our knowledge of nitrate sensing and the signaling transduction pathways has been continuously improving. The primary gap in nitrate signaling research lies in understanding cell-type-specific and tissue-specific nitrate signaling and the plant’s response to different concentrations of nitrate. In the future, single-cell omics and other cell-specific and tissue-specific methods as well as in situ and live imaging analytical approaches are needed to understand the spatial and temporal complexities of the dynamic molecular responses to nitrate. In addition, epigenetic mechanisms such as chromatin regulators and regulatory non-coding RNAs might open up new avenues in the study of nitrate signaling.

Emerging evidence suggests that under stressful conditions such as drought, salinity, or extreme temperatures, nitrate signaling pathways undergo intricate adjustments to ensure plant survival. During stress situations, nitrate availability may be limited due to reduced uptake or impaired transport within the plant. This triggers a series of adaptive responses aimed at maintaining nitrogen homeostasis and optimizing resource allocation. Plants prioritize the redistribution of nitrogen resources from older tissues to more vulnerable or actively growing parts to sustain essential processes. Nitrate influences the expression of stress-responsive genes involved in osmotic adjustment, antioxidant defense mechanisms, heat shock proteins’ production, and other adaptive responses. Nitrate cross-talks with ABA that regulate stomatal closure during drought stress. Nitrate also modulates ROS production and scavenging systems to counteract oxidative damage caused by various stressors. These results highlight the importance of nitrate in orchestrating an integrated network of molecular events that enable plants to withstand adverse conditions. Plants’ response to nitrate and abiotic stimuli may converge at certain regulatory nodes, such as phosphorylating NRT1.1 at Thr101, nucleocytoplasmic translocation of NLP7, and ROS accumulation. Identifying the underlying molecular mechanism remains an exciting challenge.

In conclusion, nitrate signaling plays a pivotal role in plant growth, resilience, and stress response. It serves as a molecular hub that integrates various physiological and molecular processes to enable plants to find balance in the face of adversity. Understanding the intricate mechanisms underlying nitrate signaling during the stress response is crucial for developing strategies to enhance crop resilience and productivity. Manipulating genes involved in nitrate uptake, transporters regulation, or downstream signaling components could potentially improve plant tolerance to multiple stress factors. Moreover, optimizing nutrient management practices and providing a balanced nitrate supply tailored to specific stress conditions may be key in maximizing plant growth and adaptation in challenging environments.

## Figures and Tables

**Figure 1 ijms-24-14406-f001:**
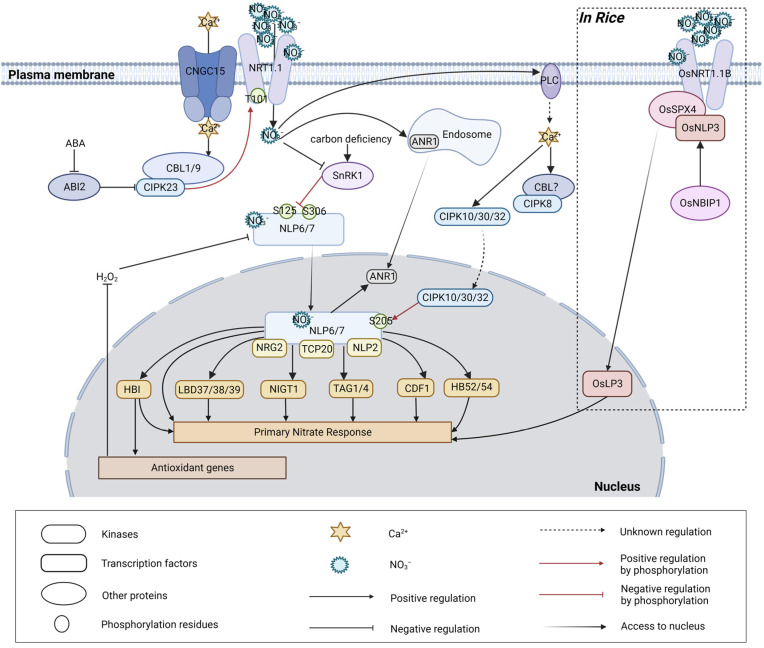
Nitrate Signaling Pathway. The dual-affinity nitrate transporter NRT1.1 is a nitrate sensor or transceptor on the plasma membrane. When the external nitrate concentration increases, the interaction between NRT1.1 and CNGC15 weakens, allowing CNGC15 to regain its calcium channel activity. PLC activity also plays a role in increasing cytoplasmic calcium concentration under nitrate provision. The low-nitrate concentration-triggered cytoplasmic calcium waves could be sensed by CBL1/9, which interacts with and activates CIPK23 to phosphorylate NRT1.1 at its Thr101 residue, switching the transporter from low-affinity to high-affinity mode. ABI2 could dephosphorylate and inactivate CBL1 and CIPK23. The activity of ABI2 is inhibited by ABA. CIPK8 is found to positively regulate the low-affinity nitrate response. The CBL protein partner and direct target(s) of CIPK8 remain to be revealed. The nitrate-triggered cytoplasmic calcium waves could be sensed by CPK10/30/32. The activated CPK10/30/32 phosphorylate NLP7 (and NLP6) at the conserved Ser205 residue to ensure its nuclear retention. Under carbon deficiency, SnRK1 phosphorylates NLP7 at Ser125 and Ser306, which increases the cytoplasmic localization and promotes the subsequent degradation of NLP7. Nitrate could release the inhibitory effects of SnRK1 by promoting its degradation. As another nitrate sensor, NLP7 directly binds to the promoters of the primary nitrate response genes and many downstream transcription factors genes. These secondary transcription factors work together with NLP7 to amplify the nitrate-response transcriptional cascade. Some transcription factors, such as NRG2 and TCP20, have direct interaction with NLP7. NLP2 and NLP6 also interact with NLP7. HBIs not only regulate PNR gene expression but also increase the expression levels of a set of antioxidant genes to reduce the accumulation of H_2_O_2_, which inhibits the nitrate-induced nuclear localization of NLP6 and NLP7, thereby forming a feedback regulatory loop to enhance the nitrate signaling. ANR1 is induced from the endosome by the Ca^2+^–CPKs–NLPs signaling pathway under high nitrate conditions. In rice, OsNRT1.1B, the postulated orthologue of the *Arabidopsis* NRT1.1 transceptor, interacts with the phosphate signaling repressor protein OsSPX4, which suppresses the activity of the master transcription factor OsNLP3. Once nitrate is sensed by OsNRT1.1B, OsNBIP1 is recruited to OsNRT1.1B–OsSPX4 complex to mediate the ubiquitination and degradation of OsSPX4. Hence, OsNLP3 is released from interaction with OsSPX4 and translocates to the nucleus to trigger the nitrate response. It is still unknown whether the NRT1.1–SPX–NLP module exists in *Arabidopsis*.

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
