# Peer review of "Finding Balance in Adversity: Nitrate Signaling as the Key to Plant Growth, Resilience, and Stress Response"

_ijms, 2023, doi:10.3390/ijms241914406_

Round 1

Reviewer 1 Report

The manuscript entitled “Nitrate Signaling in Plant Response to Abiotic Stresses” is well written. However, few things needed to add in manuscript to make it more attractive for the readers.

The manuscript contained lots of information, but need to add few tables to make easier for the readers. Tables should contain, which abiotic factors affecting NRT1.1, NLP7, and CIPK23 in which plants.

Suddenly abbreviation appeared in Line no. 169-179? Shift all the abbreviation at figure 1 main heading.

Reviewer 2 Report

The review article "Nitrate Signaling in Plant Response to Abiotic Stresses" is a sticking topic and has great potential to be reviewed. In this article, I have observed some serious concerns. The title is too general, I would recommend please re-write the title with more specific keywords.  The article looks more summary of the previous research, not a review. Arrange the headings in a proper sequence. Start with little details about abiotic stresses and plant response, then move to nitrate pathway overview and take this to NRT1.1 and so on. The perspective is very little, try to increase the perspective and future directions in more detail. I would recommend adding a detailed table about the nitrate signaling observed in different plants under various stresses. In the last, I would recommend not using old references (Not older than 2015 or 2016)

Minor changes

Round 2

Reviewer 1 Report

Authors have incorporated all the comments.  I recommend this manuscript to publish in current version.